# Molecular imaging with engineered physiology

Mitul Desai[1,*], Adrian L. Slusarczyk[1,*], Ashley Chapin[1], Mariya Barch[1] & Alan Jasanoff[1,2,3]

*In vivo* imaging techniques are powerful tools for evaluating biological systems. Relating image signals to precise molecular phenomena can be challenging, however, due to limitations of the existing optical, magnetic and radioactive imaging probe mechanisms. Here we demonstrate a concept for molecular imaging which bypasses the need for conventional imaging agents by perturbing the endogenous multimodal contrast provided by the vasculature. Variants of the calcitonin gene-related peptide artificially activate vasodilation pathways in rat brain and induce contrast changes that are readily measured by optical and magnetic resonance imaging. CGRP-based agents induce effects at nanomolar concentrations in deep tissue and can be engineered into switchable analyte-dependent forms and genetically encoded reporters suitable for molecular imaging or cell tracking. Such artificially engineered physiological changes, therefore, provide a highly versatile means for sensitive analysis of molecular events in living organisms.

[1] Department of Biological Engineering, Massachusetts Institute of Technology, 77 Massachusetts Avenue, Room 16-561, Cambridge, Massachusetts 02139, USA. [2] Department of Brain & Cognitive Sciences, Massachusetts Institute of Technology, 77 Massachusetts Avenue, Room 16-561, Cambridge, Massachusetts 02139, USA. [3] Department of Nuclear Science & Engineering, Massachusetts Institute of Technology, 77 Massachusetts Avenue, Room 16-561, Cambridge, Massachusetts 02139, USA. * These authors contributed equally to this work. Correspondence and requests for materials should be addressed to A.J. (email: jasanoff@mit.edu).

The combination of chemical probes with noninvasive imaging-based detection uniquely enables molecular and cellular phenomena to be mapped across large regions of living tissue[1]. Molecular imaging agents enable *in vivo* monitoring of a wide variety of discrete targets, such as low-molecular weight analytes, protein receptors, enzymes and metabolic processes. Applications range from basic biological investigations to clinical evaluation of diseases, and the use of molecular imaging techniques can be particularly important for analysis of hard-to-reach structures such as the brain.

Despite impressive progress, however, current molecular imaging approaches remain limited by the available forms of imaging probes. Radioactive probes are the current gold standard for clinical molecular imaging, both in brain and in peripheral tissue. These agents are detected with subnanomolar sensitivity but relatively poor spatial resolution by positron emission tomography (PET) and single photon computed tomography. Although radiotracers have enabled a host of groundbreaking imaging applications, they cannot be switched on and off by targets or analytes. This increases their toxicity, rules out biosensing approaches, and limits most dynamic measurements to pharmacokinetic time scales on the order of minutes. Probes for dynamic molecular magnetic resonance imaging (MRI) can be imaged in animal brains at submillimetre resolution[2]. Their potency can be dynamically modulated, but they are difficult to detect with sensitivity appropriate for most biological targets of interest, in the nanomolar range[3]. Techniques such as nuclear hyperpolarization[4] and chemical exchange saturation transfer[5] have been explored to boost the efficiency of MRI probe detection; in some instances, these have been reported to provide sensitivity to picomolar-scale targets *in vitro*[6]. Micron-sized MRI contrast agents[7] have been applied at comparable concentrations in live animal brains, but have so far been demonstrated primarily for cell tracking. Ultrasound and diffuse optical molecular imaging techniques also show promise[8,9], but are still in early stages of development. The need for continuing innovation of sensitive and versatile molecular imaging strategies, therefore, persists.

To address this need, we sought to establish a molecular imaging paradigm that can be applied in the brain and that bypasses limitations of existing imaging agents by manipulating a strong endogenous physiological contrast source rather than employing exogenous agents that are themselves directly measured. The vasculature is one of the most potent endogenous contrast sources available to imaging modalities. Changes in blood volume, flow or oxygenation are robustly detectable by MRI, optical and ultrasound-based noninvasive imaging, as well as by PET, single photon computed tomography and X-ray imaging performed in conjunction with intravascular tracers. The vasculature densely permeates most tissues. In the cerebral cortex, the mean distance from any point to a blood vessel is about 13 µm (ref. 10). Vascular haemodynamic changes can be evoked by a variety of chemical species, many of which act at nanomolar concentrations.

We reasoned that a strategy for 'hijacking' vascular physiology to report on specific molecular or cellular events of interest by means of engineered vasoactive probes could provide a sensitive new platform for molecular imaging. In this paper, we demonstrate that this approach is feasible and enables the construction of both analyte-responsive molecular imaging agents and genetically encodable reporters. Such molecular imaging methods based on engineering of physiological contrast sources could offer a versatile alternative to conventional contrast agents in scientific or clinical contexts.

## Results

**Platform for vasoactive imaging agents**. To demonstrate the engineered haemodynamic imaging approach, we chose to engineer calcitonin gene-related peptide (CGRP), the most potent known human vasodilatory peptide[11]. Endogenous CGRP occurs as two closely-related 37 residue isoforms that act on widely expressed RAMP1 (receptor activity-modifying protein 1)/CLR (calcitonin receptor-like receptor) receptor heterodimers to induce dilation of intracerebral arterioles with a half-effective concentration ($EC_{50}$) below 10 nM (ref. 12). This suggests that CGRP variants should be capable of artificially inducing equivalents to the well-known blood oxygen level dependent (BOLD) MRI response[13] that constitutes the basis of most functional brain imaging experiments (Fig. 1a). Molecular imaging of arbitrary analytes could be possible if activation of an initially blocked or caged CGRP analog could be coupled to presence of the analyte (Fig. 1b). Finally, because CGRP is a peptide, it could potentially be harnessed as a genetic reporter in cells and tissues (Fig. 1c). CGRP, therefore, provides an excellent basis for demonstrating the application of vasoactive molecules to several major problems in molecular imaging.

**Vasoprobe detection by *in vivo* optical imaging and MRI**. To determine the feasibility of imaging the action of CGRP, we first characterized *in vivo* vasodilation effects of wild-type human α-isoform CGRP (wtCGRP) in the rat brain using optical imaging and MRI. Optical imaging via a cranial window (Fig. 2a) permitted direct visualization of the effects of dilation of both macroscopic and microscopic blood vessels upon topical infusion of wtCGRP (0.3 ml, 50 nM) onto exposed cortex. Dilation of macroscopic vessels was quantified as percent change in vascular diameter (Fig. 2b red bar and Fig. 2c top panel), and apparent parenchymal microcapillary blood volume changes were tracked by monitoring optical reflectance, assumed to vary inversely with parenchymal blood volume, in a region of interest (Fig. 2b black box and Fig. 2c bottom). Significant increases in macroscopic vessel diameter ($9.5 \pm 2.4\%$ dilation, *t*-test $P < 10^{-9}$, $n = 5$) and decreases in optical reflectance from parenchymal tissue ($8.0 \pm 2.8\%$ intensity change, *t*-test $P < 10^{-11}$, $n = 5$) were observed following wtCGRP superfusion, demonstrating that CGRP-induced responses can be sensitively detected by *in vivo* optical imaging.

We next investigated the ability of wtCGRP to induce contrast detectable by noninvasive imaging in optically inaccessible brain regions by acquiring MRI scans during intracranial infusion of 100 nM wtCGRP in artificial cerebrospinal fluid (aCSF) into deep parenchymal tissue. Injections of wtCGRP or vehicle control at 0.1 µl min$^{-1}$ were performed through cannulae bilaterally implanted in arbitrarily selected thalamic regions of live rat brains, and MRI data were acquired using weighting parameters suitable for detecting BOLD contrast (Fig. 2d). Voxels clustered around the sites of wtCGRP injection showed substantial signal changes that were significantly correlated with a 10 min infusion epoch (corrected *t*-test $P < 0.05$, $n = 6$), whereas no voxels near control injection sites showed significant correlation. Data from six animals revealed mean peak MRI signal changes of $5.3 \pm 0.9\%$ during wtCGRP injection, compared with changes of $1.2 \pm 0.1\%$ during injection of aCSF alone; the difference between wtCGRP and vehicle was highly significant (*t*-test $P = 0.001$). Time courses of mean MRI signal changes observed during infusion were consistent with the optical imaging data and indicated that wtCGRP is not rapidly removed or degraded in brain tissue (Supplementary Fig. 1a). Haemodynamic contrast provided sufficient resolution to monitor convective spread of the imaging agent from the site of infusion (Supplementary Fig. 1b).

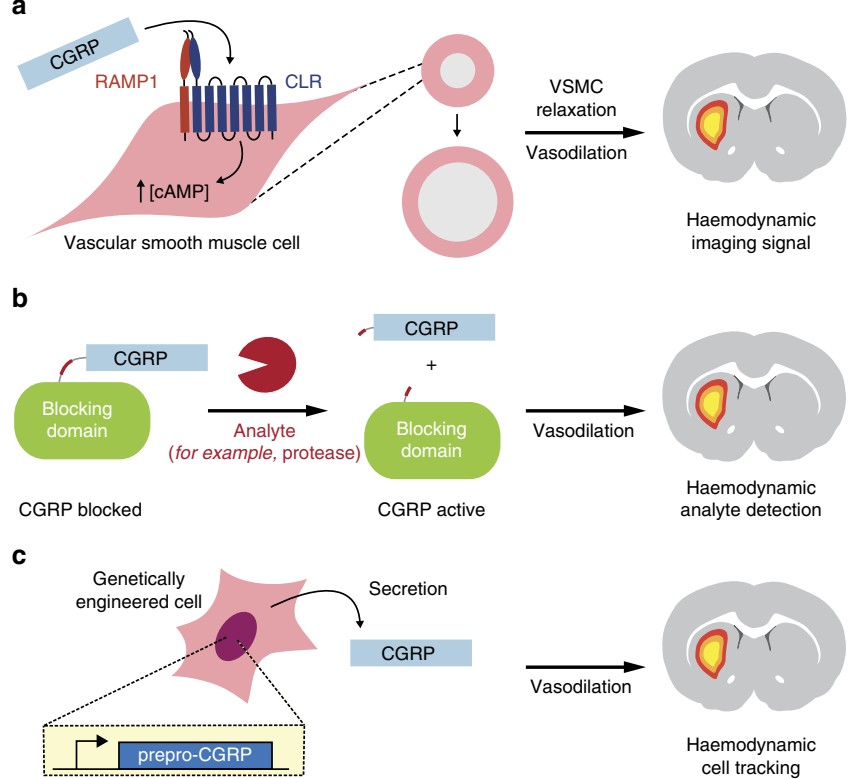

**Figure 1 | Principle of molecular imaging with CGRP-induced haemodynamic responses.** (**a**) CGRP acts on the heterodimeric G-protein coupled receptor RAMP1/CLR (left) to induce intracellular cAMP production, resulting in relaxation of vascular smooth muscle cells (VSMCs) and consequent vasodilation (middle). Dilation of microvasculature induces haemodynamic effects visible by MRI and other imaging methods (right). (**b**) Design for analyte-sensitive vasoactive probes comprising CGRP fused to a labile blocking domain via an analyte-responsive linker. Such probes can be activated by analytes that act sterically on the sensor to unblock the CGRP moiety, for example, by cleaving the blocking domain as shown here. This would induce an analyte-dependent haemodynamic response and enable molecular imaging with CGRP-based constructs. (**c**) Application of CGRP as a genetically encoded reporter would involve expressing prepro-CGRP in genetically modified cells or tissues. Subsequent processing of the construct should result in secreted CGRP, allowing the genetically modified cells to be tracked by haemodynamic imaging methods.

To dissociate wtCGRP-induced image contrast from haemodynamic effects evoked via pathways associated with neural activity, we performed wtCGRP infusion in the presence or absence of neuronal nitric oxide synthase (nNOS) inhibition, which has previously been shown to reduce neural stimulation-evoked BOLD signals by over 60% (ref. 14). MRI response profiles and amplitudes (Fig. 2e) evoked by wtCGRP, before and after systemic intravenous injection of the nNOS inhibitor $N$-nitro-L-arginine methyl ester (L-NAME; $50\,mg\,kg^{-1}$) (ref. 15) were statistically indistinguishable ($t$-test $P > 0.05$). Although L-NAME injection itself appeared to induce a small average increase in the baseline brain signal before CGRP infusion ($3.5 \pm 2.3\%$), this change was not statistically significant ($t$-test $P = 0.11$, $n = 4$). These results demonstrate that wtCGRP can be used to engineer substantial and specific image contrast changes in deep brain, at concentrations about 1,000-fold lower than those at which conventional MRI contrast agents produce equivalent effects[2].

**Design of vasoactive protease sensors.** To perform molecular imaging of biochemical targets using CGRP-mediated contrast, we implemented an analyte-dependent uncaging scheme as shown in Fig. 1b. We focused on detection of proteases, which are important as drug targets, diagnostic biomarkers and bases for biotechnological applications such as cell signalling and prodrug activation[16,17]. Since proteases are subject to extensive post-translational regulation, it is often important to measure their activity rather than their abundance[18], thus necessitating enzymatically sensitive probes *per se*.

To rapidly evaluate candidate CGRP-based protease activity sensors, we developed an *in vitro* bioassay for CGRP receptor activation in cultured cells (Supplementary Fig. 2). Cells used for the bioassay were engineered to co-express the RAMP1/CLR complex in conjunction with a bioluminescent reporter sensitive to RAMP1/CLR-mediated cyclic adenosine monophosphate (cAMP) production, such that effective receptor activation leads to luminescence signals readily detected in a microtiter plate reader. Measurements using this system indicated an $EC_{50}$ for receptor activation by wtCGRP of 48 pM (95% confidence interval (CI) 37–63 pM), which was 2.5- to 14-fold impaired by N-terminal extensions of one to three residues (Supplementary Fig. 3). C-terminal extension of wtCGRP by addition of a glycine residue in contrast resulted in a much higher $EC_{50}$ of 2.5 nM (CI 2.1–3.1 nM), and substitution of the C-terminal amide of wtCGRP by a carboxylate reduced potency of the peptide by a factor of ~250. These findings are consistent with earlier studies of terminal modifications of CGRP[19]. The relatively modest effects of N-terminal modifications to wtCGRP suggested that protease sensors might best be formed by fusing blocking domains via protease cleavage sites to the N terminus of CGRP, so that cleavage products could recover vasodilation activity close to the wild-type variant.

Guided by these findings, we designed candidate vasoactive sensors for a number of proteases relevant to biotechnology (Fig. 3a). Targets included fibroblast activation protein (FAP), a cell surface-bound collagenase and cancer biomarker[20], which removes the dipeptide AP from the N terminus of polypeptides;

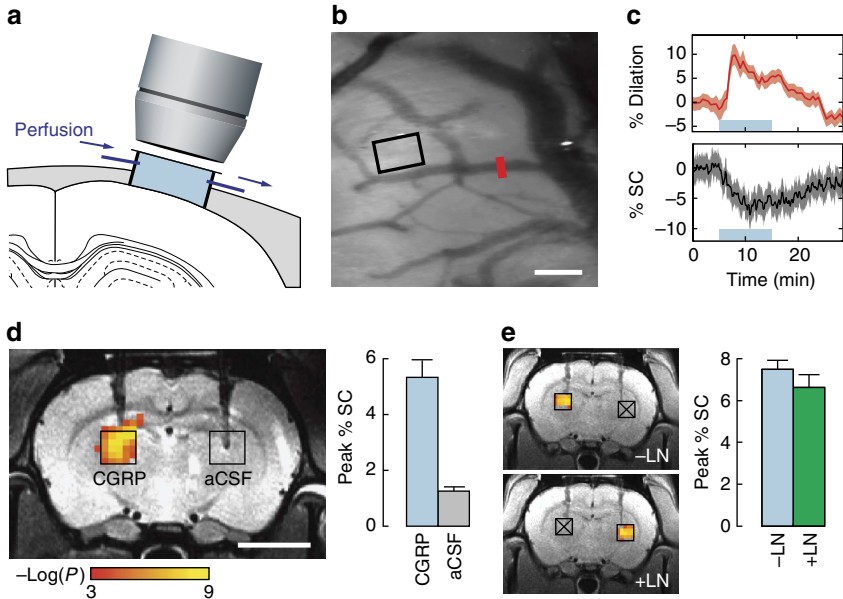

**Figure 2 | Imaging artificial haemodynamic responses induced by CGRP probes *in vivo*.** (**a**) Direct effects of superficially perfused CGRP on cerebral vasculature are assessed by optical reflectance imaging of exposed cortex. (**b**) Cortical image of a single rat, revealing discrete blood vessels (dark) and parenchymal tissue (light background). Scale bar, 100 μm. (**c**) Time course of average CGRP-induced haemodynamic changes in five animals; shading denotes standard errors as a function of time. The CGRP infusion period is denoted by the blue shaded lines. Top panel shows changes in the diameter of macroscopic vessels as indicated by the red bar in **b**. Bottom panel shows mean optical signal changes in parenchymal regions as denoted by the black rectangle in **b**; downward deflection of the curve during CGRP administration is consistent with increasing parenchymal cerebral blood volume, which darkens images. (**d**) MRI changes observed in conjunction with deep intracranial injection of CGRP (100 nM) or vehicle only (aCSF), both at 0.1 μl min$^{-1}$. for 10 min into the parenchyma of the thalamus. Coloured voxels overlaid on a greyscale anatomical image (left) indicate the correlation strength of observed signal changes with the injection period. Bar graph (right) indicates peak signal changes observed upon injection of CGRP or aCSF, each averaged over corresponding injection regions in multiple animals (black boxes at left, $n = 6$). Scale bar, 5 mm. (**e**) Dissociation of CGRP-induced artificial haemodynamic signals from nitric oxide-mediated signalling. Left, haemodynamic responses observed by imaging upon unilateral CGRP injection into the left thalamus, before L-NAME delivery (-LN, top), or into the right thalamus, after 50 mg kg$^{-1}$ intravenous L-NAME infusion (+LN, bottom). Sites denoted by crossed out boxes did not receive CGRP injections under the conditions labelled. Right, comparison of peak percent signal changes observed in regions as identified on the left, without and with L-NAME and showing no significant difference. Error bars throughout this figure denote s.e.m. values ($n = 4$).

TEV protease and enterokinase, both well-studied endopeptidases that are absent in the central nervous system and could, therefore, be used as bio-orthogonal markers for transgenesis or cell tracking[21,22]; and caspase-3 (CASP3), another endopeptidease and cytosolic apoptosis actuator that can be released following apoptosis and secondary necrosis[23] and serve as a prognostic indicator during cancer treatments[24,25]. FAP and CASP3 have been targeted by previous optical[26–28] or MRI[29,30] probes, but improvements in sensitivity for deep tissue detection would be valuable. CGRP constructs targeted towards these proteases and incorporating AP repeats, green fluorescent protein (GFP), or biotin as blocking domains were produced in *Escherichia coli* or by solid phase peptide synthesis.

**Experimental validation of vasoactive protease sensors.** We then used our *in vitro* bioassay to identify protease sensor candidates that displayed minimal activity before cleavage and high potency after enzyme treatment. We found that uncleaved recombinant sensors of the format GFP-(cleavable linker)-CGRP-G were completely inactive as CGRP receptor agonists even at concentrations of 10 μM (Fig. 3b and Supplementary Fig. 4). After 2 h incubation with their cognate proteases (TEV, enterokinase or CASP3), the constructs were largely cleaved and elicited bioluminescence in RAMP1/CLR reporter cells with apparent EC$_{50}$ values from ~0.4–1.0 μM (Table 1). Amidated synthetic CGRP-based substrates for FAP and CASP3 regained much higher levels of potency following cleavage, with EC$_{50}$

values for FAP of 16 pM (CI 13–20 pM) and for CASP3 of 42 pM (CI 38–46 pM). These constructs were less effectively inactivated by blocking domains before cleavage, however, with EC$_{50}$ values of 114 pM (CI 112–116 pM) and 5.7 nM (CI 5.2–6.4 nM) for FAP and CASP3, respectively. Effective conditions for protease sensing by each of the CGRP constructs could be identified based on the shift in EC$_{50}$ values measured in the bioassay upon cleavage (Fig. 3c), indicating that each of the candidates could potentially function as a molecular imaging probe for protease activity *in vivo*. Because the optimum protease sensitivity and post-cleavage potency was displayed by the biotin-(CASP3 site)-CGRP construct, this variant was selected for validation studies in animals.

To demonstrate imaging-based detection of the difference between activated and unactivated CASP3 probes, we again used the intracranial injection procedure of Fig. 2c. CASP3 sensor aliquots (100 nM) were injected at 0.1 μl min$^{-1}$ for 10 min in the presence or absence of CASP3 enzyme (2.3 ng μl$^{-1}$) in paired injections into rat thalamus. MRI scans acquired with BOLD contrast weighting revealed clusters of voxels for which significant correlation between the injected sensor and the image signal (corrected *t*-test $P < 0.05$) was observed in the presence but not the absence of the co-injected protease (Fig. 3d). The mean MRI peak signal change observed during sensor/CASP3 co-injection was $7.0 \pm 0.9\%$, whereas the peak signal change observed in the absence of CASP3 was only $1.6 \pm 0.5\%$; this difference was highly significant, with *t*-test $P = 0.02$ ($n = 4$). These results demonstrate that CGRP-based CASP3 sensors elicit engineered haemodynamic

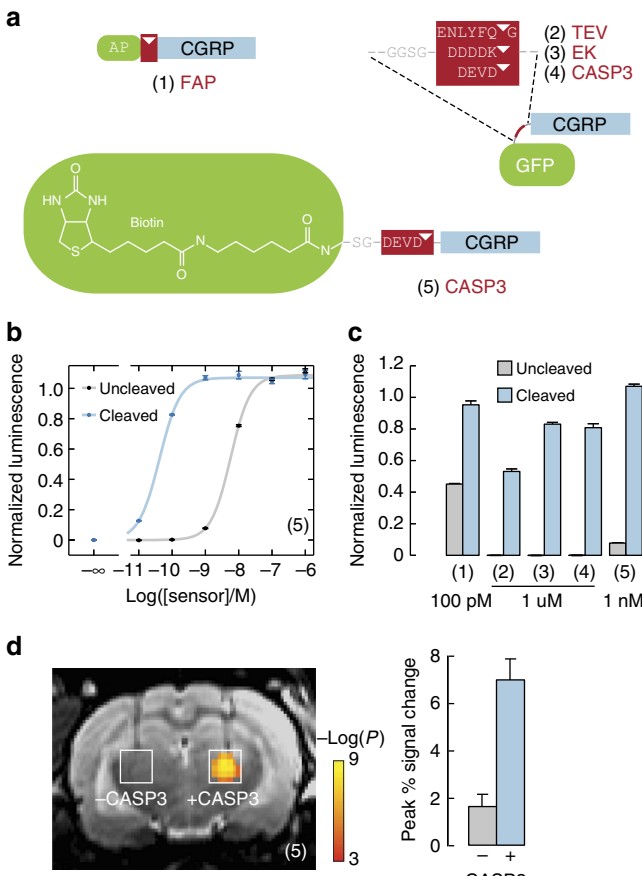

**Figure 3 | Design and assessment of protease-activated imaging probes based on CGRP.** (**a**) Protease sensor designs incorporate an N-terminal blocking moiety (green), a protease cleavage sequence (red, cleavage sites indicated by triangles), and a C-terminal CGRP moiety (blue). Structures of five protease sensors are shown, with cognate proteases labelled in red text: (1) The synthetic peptide AP-CGRP-amide detects dipeptidase activity of fibroblast activation protein (FAP). (2–4) Recombinant fusion proteins comprises cysteine-free GFP, a short linker, a protease site and non-amidated CGRP detect TEV protease (2), enterokinase (EK) (3) and caspase-3 (CASP3) (4). (5) The synthetic peptide (long-chain biotin)-SG-DEVD-CGRP-amide also detects CASP3 activity. (**b**) Dose–response curve for sensor (5) measured using a cell-based bioassay[32] (Supplementary Fig. 2) following incubation with or without CASP3. Luminescence values were normalized, and error bars reflect s.d. from $n = 3$ replicates. (**c**) Protease sensing by CGRP-based probes measured *in vitro*, using probe concentrations indicated. Error bars represent s.d. of $n = 3$ replicates. Sensors were incubated with or without corresponding proteases: (1) with 5 ng µl$^{-1}$ of human FAP; (2) with 0.1 U µl$^{-1}$ of TEV protease; (3) with 2 pg µl$^{-1}$ of EK light chain; (4) with 23 ng ul$^{-1}$ and (5) with 11.5 ng µl$^{-1}$ of human CASP3. (**d**) Left, protease-dependent switching of 100 nM CGRP-based molecular imaging probe (5) induces contrast differences in MRI. Significant haemodynamic activation can be seen in the presence but not the absence of coadministered 1.15 ng µl$^{-1}$ CASP3. Right, bar graph showing peak signal change induced by uncleaved versus cleaved sensor (5). Error bars throughout this figure indicate s.e.m. values over $n = 4$ animals.

contrast differences appropriate for noninvasive molecular imaging in deep brain regions. The volumes over which probe-dependent responses were observed, as well as experience with conventional MRI contrast agent injections[2], suggest that CGRP variants were diluted approximately tenfold from 100 nM levels in the infusion cannulae to effective concentrations near

10 nM in the brain parenchyma. The magnitude of observed MRI changes elicited by these probe levels suggests that substantially lower CGRP concentrations would also be detectable in the brain.

**Construction and demonstration of a vasoactive genetic reporter.** In addition to their utility as building blocks for molecular sensors, peptidic vasoactive agents like CGRP could function as reporters of transgene expression. As an initial test of this idea, we sought to determine whether xenografted cells genetically modified to secrete CGRP could be visualized by imaging in rat brain, as schematized in Fig. 1c. Endogenous CGRP is produced by proteolytic processing of a prepropeptide and export by machinery common to mammalian cells. To exploit this mechanism artificially, we constructed a bicistronic vector encoding prepro-CGRP as well as the fluorescent protein mKate linked to the ampicillin resistance gene Bla via a self-cleaving 2A peptide (Fig. 4a).

HEK293FT cells were transfected with the prepro-CGRP/mKate-2A-Bla construct or with a control vector encoding only mKate-2A-Bla and cultured on selective medium to obtain stable lines. CGRP production was then assessed by washing the cells and then performing the RAMP1/CLR activation bioassay of Supplementary Fig. 2 on aliquots of growth medium withdrawn at defined time intervals following the wash. Results were compared with a standard curve obtained with known amounts of synthetic CGRP and demonstrated that cells expressing prepro-CGRP release an average of between $10^{-19}$ and $10^{-18}$ mol CGRP per cell over a 24 h period. Given the detection limit for CGRP below 10 nM by MRI in rat brain, we, therefore, predicted that cell densities between $10^4$ and $10^5$ cells per µl should be easily detectable over analogous time periods *in vivo*. This is competitive with existing genetic MRI reporters for cell tracing[21] and could be further optimized if desired.

To assess the ability of CGRP-expression constructs to induce detectable vascular contrast *in vivo*, cells expressing the prepro-CGRP/mKate-2A-Bla vector or the control vector were injected intracranially into rat striatum and imaged by MRI at time points immediately before and 1 day after implantation. Signal brightening consistent with vasodilation in the neighbourhood of the CGRP cell injection, but not the control injection, was discernable by simple visual inspection of $T_2$-weighted MRI scans obtained at the 24 h time point (Fig. 4b). Quantitative analysis of the signal change observed between day 0 and day 1 (Fig. 4c) revealed an average of $23 \pm 3\%$ signal change induced in the region of CGRP-expressing cells and an average of $7 \pm 2\%$ signal change induced near control cells; this difference was highly significant (t-test $P = 0.01$, $n = 5$), demonstrated specific detection of the CGRP expression construct. The larger magnitude of observed signal change, compared with effects of acute CGRP injection (Fig. 2c), may reflect a greater extent of affected tissue. Importantly, the region of MRI signal enhancement observed in individual animals corresponded closely to the distribution of transplanted prepro-CGRP-expressing cells, as indicated by post-mortem histological analysis of mKate fluorescence after MRI procedures (Fig. 4d). Injected control cells displayed robust mKate expression in the absence of MRI enhancements, and neither test nor control injections resulted in gross tissue disruption or evidence of toxicity (Supplementary Fig. 5). These results, therefore, confirm the utility of CGRP as a genetic tool for cell monitoring in the brain. Importantly, applications can be envisioned which combine genetic control of vasoactive agents with analyte-switchable behaviour to probe both internal cell state and the extracellular biochemical context of genetically specified cell populations, a key problem in the field[31].

## Discussion

Our data show that engineered haemodynamic signals can provide a sensitive, highly versatile, and conceptually unprecedented alternative to established molecular imaging approaches. By exploiting changes in an endogenous physiological property, this strategy avoids the need for directly detected imaging agents. Concentrations of vasoactive molecules necessary for imaging approach the low tracer doses used in nuclear medicine techniques like fluorodeoxyglucose PET[32], and unlike radiotracers, CGRP derivatives are activatable by molecular targets, genetically encodable, intrinsically less hazardous, detectable at higher spatiotemporal resolution, and compatible with multiple diagnostic modalities. Utility of the new probe technology for genetic reporting in the brain is directly demonstrated by our methods, and can enable tracking of genetically labelled transplanted cells, monitoring of gene therapy or detection of genetic program activation in engineered cells and tissues. Vasoactive protease sensors introduced here only constitute a proof-of-concept for analyte-dependent molecular imaging, but could be adapted and applied in future work for measurement of a large variety of biological targets of interest.

Further work can improve in a number of ways on the new technology introduced here. Improvements in probe delivery are especially critical. Application of vasoactive probes beyond the brain should benefit from widespread activity of CGRP in multiple tissues[33], but has yet to be demonstrated. For future non-genetic applications, development of a strategy that promotes efficient extravasation of intravenously injected CGRP-based probes might be critical for molecular imaging in some tissues. In the brain, noninvasive delivery past the blood–brain barrier (BBB) will be a particular priority, especially to address the challenge of translating this method into human subjects. Fortunately, the minimal amounts of vasoactive probes required for engineered haemodynamic imaging are likely to be conducive in the future to noninvasive trans-BBB delivery by means of receptor-mediated transcytosis, a delivery technique which is proving successful in rodents, non-human primates and human clinical trials[34]. A variety of alternative vector-based or BBB disruption-dependent strategies might also enable effective brain delivery[35], particularly given the low concentrations required for molecular imaging with vasoactive agents.

**Table 1 | Effect of cleavage on the potency of CGRP-based protease sensors.**

| Construct | Log(EC$_{50}$) | |
|---|---|---|
| | Uncleaved | Cleaved |
| CGRP-NH$_2$ | NA | −10.3 |
| CGRP-G | NA | −8.6 |
| Sensor (1) | −9.9 | −10.8 |
| Sensor (2) | < −5 | −6.0 |
| Sensor (3) | < −5 | −6.4 |
| Sensor (4) | < −5 | −6.4 |
| Sensor (5) | −8.2 | −10.4 |

NA, not applicable.
Log(EC$_{50}$) values obtained by dose–response curve measurement from sensors (1–5) from Fig. 3 incubated with or without corresponding proteases: (1) with 5 ng μl$^{-1}$ of human FAP; (2) with 0.1 U μl$^{-1}$ of TEV protease; (3) with 2 pg μl$^{-1}$ of enterokinase light chain; (4) with 23 ng μl$^{-1}$ and (5) with 11.5 ng μl$^{-1}$ of human CASP3. s.e.m. values were all <0.1 (n = 7).

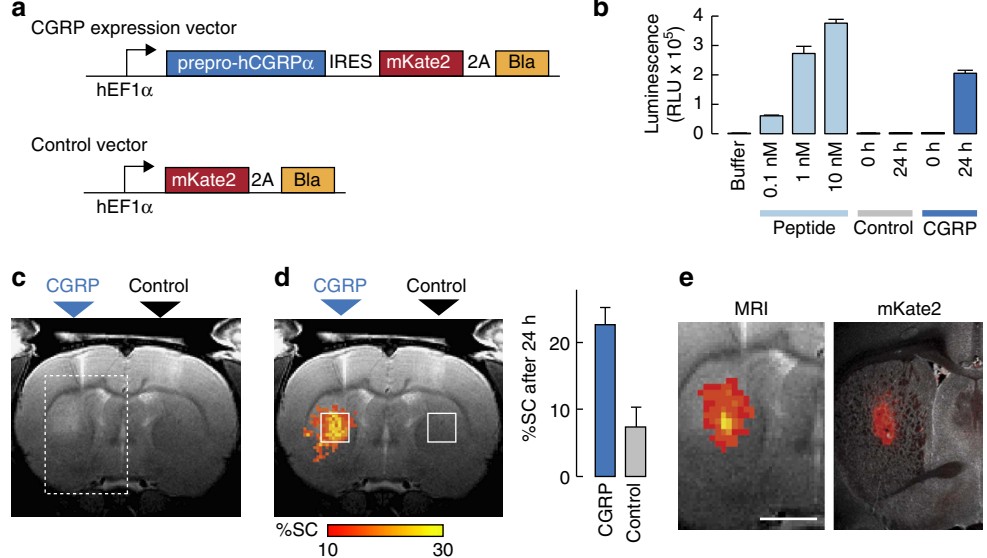

**Figure 4 | Application of CGRP as a genetically expressed vasoactive reporter.** (**a**) Lentiviral vectors were used for expression of prepro-CGRP or a control construct. Top: probe vector controlled by the hEF1α promoter, encoding human prepro-CGRP (prepro-hCGRPα), an internal ribosome entry site (IRES), and the red fluorescent protein mKate2 joined by a self-cleaving viral 2A peptide to the blasticidin selection marker (Bla). Bottom: control vector lacking prepro-hCGRPα and the IRES sequence. (**b**) In vitro demonstration of substantial CGRP release from transfected HEK293FT cells. RAMP1/CLR activation by aliquots of purified CGRP peptide (light blue) or supernatants from cultured cells expressing control (grey) and prepro-CGRP (dark blue) lentiviral constructs was measured using the bioassay of Supplementary Fig. 2. Cell supernatants were assayed at 0 and 24 h after an initial washing, and all samples were assayed at tenfold dilution. Error bars denote s.d. of n = 3 replicates. (**c**) In vivo detection of implanted HEK293FT cells producing CGRP in rat brains. A representative $T_2$-weighted MRI scan obtained 24 h after striatal injection of 3 × 10$^5$ CGRP-producing or control cells (labels above) shows discernable signal enhancement consistent with probe-induced vasodilation in the CGRP injection region (dashed box) but not near the control injection site opposite. (**d**) Left: group-averaged map (left) displaying percent signal change (%SC, colour scale) observed between 0 and 24 h after cell implantation, averaged over five animals. Right: mean %SC in the square regions of interest superimposed on the image. Error bars denote s.e.m. across five animals. (**e**) Correspondence of MRI signal change (left, colour scale as in **d**) measured in vivo and mKate expression (right, red superimposed over a bright field image) visualized postmortem in a representative animal. The field of view corresponds to the rectangular box in **c**. Scale bar, 2 mm.

Additional limitations of vasoactive probe technology must also be addressed in future research. One problem is the semi-quantitative nature of molecular imaging with vasoactive probes, compared with approaches based on more conventional imaging agents. This limitation arises from the nonlinear relationships among vasoactive probe concentration, haemodynamic parameters, and contrast patterns produced in images[36]. Although quantification of probe responses might not be required for detecting the mere presence versus absence of analytes or probe-expressing cells, more quantitative readouts would be needed for absolute measurements of continuously varying molecular or cellular parameters, such as analyte concentrations, or for comparison of such variables across different subjects or treatments. In these cases, approximately quantitative responses would have to be inferred from *in vitro* probe measurements (EC$_{50}$ values and receptor affinities), in combination with haemodynamic modelling[37,38] and calibration techniques[39,40]. In animals, it might also be possible to quantify vasoactive probe readouts with respect to standard curves obtained by injecting known concentrations of probes and analytes. A possible confound to either qualitative or quantitative vasoactive probe measurements could arise in situations where cerebral vasculature is perturbed by injury or pathology. Conditions such as ischemia[41] and haemorrhage[42] are likely to affect vascular contrast at least in MRI, and may also compromise vascular reactivity required for visualizing vasoactive probes. A further potential complication with the use of CGRP-based probes in particular might arise from the role of CGRP signalling in migraine[43,44]. This problem could be circumvented using vasoactive alternatives to CGRP, such as molecular interventions in nitric oxide, cyclooxygenase, and norepinephrine-mediated vasodilation and vasoconstriction pathways[45]. In each case, however, physiological side effects of candidate probes would need to be investigated and mitigated or controlled for where appropriate.

In addition to the possibility of performing molecular imaging using various different engineered haemodynamic response mechanisms, it might be possible to create diagnostic outputs by exploiting physiological processes unrelated to blood flow. Examples might include biological systems related to endothermia, secretion, or autonomic responses, for instance. In each case, manipulation of endogenous image contrast of physiological origin could benefit from endogenous signalling pathways and amplification mechanisms, while avoiding the adverse consequences of many exogenous diagnostic agents. Molecular imaging with engineered physiology, therefore, represents a versatile paradigm for biological experimentation and diagnostic medicine, as well as a basis for extension of 'synthetic biology' concepts to organismal scale.

## Methods

**Animal procedures.** All animal procedures were conducted in accordance with National Institutes of Health guidelines and with the approval of the MIT Committee on Animal Care. Sample sizes were estimated based on the effect size observed in previously published studies to achieve power of 0.8 and alpha = 0.05 using Biomath software (http://biomath.info/power/). All experiments were performed with male Sprague-Dawley rats, age 10–12 weeks, supplied by Charles River Laboratories (Wilmington, MA).

**Optical imaging of exposed cortex.** Optical imaging experiments were performed using Sprague-Dawley rats (300–400 g; Charles River, Wilmington, MA). To probe the impact of wild-type α-isoform calcitonin gene related peptide (wtCGRP) on vascular diameter and parenchymal blood volume *in vivo*, we measured the effects of topical application to the primary somatosensory cortex (SI) of rats. Animals (*n* = 5) were initially anaesthetized with isoflurane (3% in 50:50 air:O$_2$ mixture for induction; 1% for maintenance). Breathing rate and end-tidal expired isoflurane (V9004 Capnograph Series, Surgivet, Waukesha, WI) were continuously monitored during the imaging experiment. The animals were positioned on the surgical stereotax and the isoflurane anaesthesia was maintained

at 1% during imaging. During the surgical and imaging procedures, animals were maintained at ~37 °C core temperature using a heating blanket.

The preparation for optical imaging is diagrammed in Fig. 2a. A craniotomy (diameter of ~5 mm in rats) and durotomy were performed over primary somatosensory cortex (SI) of each animal, and the cortex was protected with Kwik-Cast silicone elastomer sealant (WPI, Sarasota, FL, USA). An imaging chamber was then attached with dental cement and the elastomer was removed. The imaging chamber was custom made using a 1 cm diameter plastic ring (McMaster-Carr, Elmhurst, IL, USA) with a base cut to match the skull topography over each rat's SI region. To facilitate wtCGRP perfusion at a constant rate, holes for fluid inlet, outlet and pressure regulation were drilled in the wall of this chamber, through which blunted 18-gauge stainless steel needles were attached and sealed using superglue. The pressure in the imaging chamber was regulated by a small vertical tube whose height could be adjusted. Inflow and outflow were controlled using two separate syringe pumps (Harvard Apparatus, Holliston, MA, USA) that were operated at the same rate (1.8 ml per h). The volume of the imaging chamber was ~0.3 ml, and this chamber was filled with artificial cerebral spinal fluid (aCSF; Harvard Apparatus, Holliston, MA, USA) and sealed with a cover glass.

A charge-coupled device camera (Prosilica GC, Allied Vision, Newburyport, MA) attached to a dissection microscope (Stemi SV11 M2 Bio, Carl Zeiss AG, Oberkochen, Germany) was used to image the cortical surface at a frame rate of ~4 Hz. Illumination was regulated using a xenon arc lamp. A green band-pass filter (550 +/− 25 nm) was used to provide optimal vascular contrast. A 50 nM wtCGRP (human α-CGRP, Sigma-Aldrich, St Louis, MO, USA) solution was prepared in aCSF for each imaging experiment. To establish imaging baselines, aCSF alone was continuously infused (1.8 ml per h) through the inlet port for 5 min before wtCGRP infusion; images were acquired continuously throughout. Fluid was extracted from the imaging chamber through the outlet port at an equal rate, in order to maintain a constant pressure. A microfluidic switch designed to minimize propulsion impact and delay with 12 μl dead space was used at the end of the baseline period to switch the infusion from aCSF to 50 nM wtCGRP in aCSF solution. A volume of 0.3 ml total of this material was perfused at the rate of 1.8 ml per hour. Impact of wtCGRP on the diameter of a cerebral artery (red bar, Fig. 2b) and on parenchymal blood volume (black rectangle, Fig. 2b) was quantified using ImageJ[46]. Image signal intensity in a region without discernable blood vessels was used as an indicator of parenchymal volume changes. Statistical analysis of all measurements was performed in Matlab (Mathworks, Natick, MA, USA).

**Magnetic resonance imaging.** Nineteen Sprague-Dawley rats were used for *in vivo* MRI experiments. Before MRI experiments, rats underwent surgery under isoflurane anaesthesia, and bilateral cannula guides (22 gauge, Plastics One, Roanoke, VA) were implanted over the ventral posterolateral nucleus of the thalamus (VPL; 3 mm lateral to midline, 4 mm posterior to bregma and a depth of 5 mm from the cortical surface) for data shown in Figs 2 and 3, or the caudate-putamen region of the striatum (CPu; 3 mm lateral to midline, 0.5 mm anterior to bregma and a depth of 5 mm from the cortical surface) for the data shown in Fig. 4. A head post was also attached atop their skulls using dental cement (C&B Metabond, Parkell, Inc., Edgewood, NY, USA), during this surgery. Cannula guides were sealed with dummy cannulae to avoid exposure of brain tissue during the recovery period. Further experiments were performed after three or more days of post-operative care. Immediately before each experiment, two injection cannulae (28 gauge, Plastics One, Roanoke, VA, USA) were attached to 25 μl Hamilton glass syringes and prefilled with the appropriate intracranial injection solution (aCSF, wtCGRP, cleaved or uncleaved CGRP-based sensors, CGRP-expressing or control HEK293FT cells). Injection cannulae were then lowered into the previously implanted bilateral cannula guides. The Hamilton syringes were then placed in a remote infuse/withdraw dual syringe pump (PHD 22/2,000; Harvard Apparatus, Holliston, MA, USA).

Animals were scanned by MRI to measure the changes in haemodynamic contrast following intracranial injections. Data were acquired on a 7 T 20 cm inner diameter, horizontal bore magnet (Bruker BioSpin MRI GmbH, Ettlingen, Germany). Home-built and commercial (Insight Neuroimaging Systems, Worcester, MA, USA & Doty Scientific, Columbia, SC) radiofrequency coils designed for rat brain imaging were used for excitation and detection. During imaging experiments, animals were anaesthetized with isoflurane (3% in oxygen for induction; 1% for maintenance). Breathing rate and end-tidal expired isoflurane were continuously monitored. Anaesthetized animals were attached via their head posts to a head holder designed to fit within the radiofrequency coil systems. Animals with their radiofrequency coils were inserted into the magnet bore and locked in a position such that the head of the animal was at the centre of the magnet bore.

**MRI assessment of injected CGRP probes and sensors.** High resolution $T_2$-weighted anatomical scans of each animal were obtained using a rapid acquisition with relaxation enhancement (RARE) pulse sequence with echo time (TE) = 44 ms, recycle time (TR) = 2,500 ms, RARE factor = 8, spatial resolution = 100 × 100 × 500 μm, and matrix size = 128 × 128 with seven slices. Haemodynamic contrast image series were acquired using a gradient echo echo planar imaging (EPI) pulse sequence with TE = 34 ms, TR = 4,000 ms, spatial resolution = 300 × 300 × 500 μm, and matrix size = 64 × 64 with seven slices. To parallel the optical imaging experiments, 5 min of baseline measurement with

4 s per image were acquired before probe infusion. Following this baseline period, while continuously collecting EPI scans, infusion pumps were remotely turned on to commence intracranial injection (in the parenchymal tissue) of 1 µl aliquots of CGRP or control solutions at the rate of 0.1 µl min$^{-1}$ through the cannulae. For the experiments of Fig. 2d, aCSF (control) and 100 nM wtCGRP in aCSF were injected simultaneously through the bilaterally implanted cannulas. For the experiments of Fig. 3d, CGRP-based CASP3 sensors (100 nM) were co-injected simultaneously, either with or without comixed CASP3 enzyme (1.15 ng µl$^{-1}$). For the nNOS inhibition experiment of Fig. 2e, a bilateral simultaneous injection of wtCGRP and aCSF was performed as in Fig. 2d. Then 50 mg kg$^{-1}$ of the nNOS inhibitor N-nitro-L-arginine methyl ester (L-NAME, Sigma-Aldrich, St Louis, MO, USA) was injected via the tail vein. Fifteen minutes after L-NAME injection, wtCGRP and aCSF injection lines were switched, and intracranially injected to assess the effect of CGRP on haemodynamic contrast in the presence of nNOS inhibition.

MRI data was processed and analysed using the AFNI software package[47]. The AFNI 3dAllineate command was used to align each animal's EPI data set to the corresponding RARE anatomical image. Each animal's image data were then aligned to a reference MRI rat atlas[48]. To identify voxels with significant increases or decreases in BOLD signal, we used the 3dDeconvolve routines in AFNI to correlate individual or group level voxel-level signal changes with the injection epochs. For correlation analysis, injection epochs were modeled by 'box car' regressors with a single stimulus block synchronized with the onset and offset of infusion, without delays. Activation in a region was deemed significant if a cluster of at least four contiguous voxels displayed a raw P-value < 0.001 for positive or negative correlation between the voxel time courses and the regressors. No significant negative correlation was observed. The cluster size and P value threshold were objectively chosen, based on AFNI's AlphaSim routine, to provide a type I error rate of 5% after correction for multiple comparisons[49,50]; this was appropriate for robust hypothesis testing on individual voxels throughout the entire imaging volume. For visualization, group statistical maps were overlaid on a reference anatomical image as in Fig. 2d. Time courses were obtained by averaging MRI signal over 1.5 × 1.5 mm regions of interest defined around cannula tip locations in individual animals' data sets. The peak percent signal change was determined by comparing signal values during baseline and infusion conditions. Additional statistical analysis was performed in Matlab.

**MRI of implanted cells.** High resolution $T_2$-weighted anatomical scans of each of animal ($n = 5$) were obtained using a RARE pulse sequence with TE = 44 ms, TR = 2,500 ms, RARE factor = 8, spatial resolution = 100 × 100 × 1,000 µm and matrix size = 128 × 128 with seven slices before injecting HEK293FT cells (day 0). After an initial scan, 3 µl of the cell suspension solution (~100,000 cells per µl) was intracranially injected in CPu (CGRP-expressing cells in left hemisphere and control cells in the right hemisphere) at the rate of 0.1 µl min$^{-1}$. Animals were taken out from the scanner, recovered form anaesthesia, and returned to their home cages.

Animals were reimaged 24 h after the cell injection (day 1) using the same scan parameters as applied before the cell injection. To quantify the signal change in the $T_2$-weighted MRI images, day 0 and day 1 data sets were registered with each other using the 3dAllineate routine in AFNI. For visualization, maps of percent signal change on day 1 compared with day 0 were arithmetically computed and a map of average changes was overlaid in colour over a greyscale anatomical image. Thresholds for display were chosen based on signal changes observed outside the regions of cell injection. Quantification of the percent signal change on day 1 compared with day 0 was performed within 1.5 × 1.5 mm square regions of interest (white squares in Fig. 4d) defined around the sites of CGRP or control cell injection.

**Histology.** After MRI cell tracking experiments, animals were transcardially perfused with phosphate buffered saline followed by 4% formaldehyde in phosphate buffered saline. Brains were extracted, post-fixed overnight at 4 °C, and then cryoprotected in 30% sucrose for 24–48 h before sectioning. Free-floating sections (50 µm) were cut using a vibratome (Leica VT1200 S, Leica Microsystems Gmbh, Wetzlar, Germany), mounted on glass slides with Vectashield mounting medium with 4,6-diamidino-2-phenylindole (Vector Laboratories, Burlingame, CA, USA) and protected with a coverslip. The distribution of injected HEK293FT cells was indicated by red fluorescence due to mKate2 reporter expression in both CGRP-producing and control cells.

**Plasmids.** A tabulated list and sequences for all plasmids used in this study are provided as Supplementary Tables 1–8. Lentiviral helper plasmids pMD2.G (Addgene #12259, Cambridge, MA) and psPAX2 (Addgene #12260) were gifts from Didier Trono. Plasmids pEF-ENTR A (Addgene #17427) and pLenti X1 Zeo (Addgene #17299) were gifts from Eric Campeau[51]. Bacterial expression plasmids Z503, Z507 and Z508 were cloned using the Golden Gate method[52] by assembling synthetic DNA sequences (IDT, Coralville, IA, USA) and PCR amplicons into the backbone of pEF-ENTR A. cfSGFP2 is a cysteine-free variant of the GFP[53]. CGRP is the alpha-isoform of human CGRP. Lentiviral plasmids were cloned using the Golden Gate method by assembling fragments for the polycistronic expression cassettes into a variant of the pLentiX1 Zeo plasmid with its kanamycin resistance

replaced by the ampicillin resistance cassette from pUC18. Each lentiviral plasmid used in this study contains a gene of interest followed by an internal ribosome entry site and a selection marker comprising a fluorescent protein, a 2A viral sequence[54], and an antibiotic resistance gene. The HA-tagged human CLR and myc-tagged human RAMP1 sequences, together encoding the two components of human CGRP receptor, were previously functionally characterized[55]. The Glo22F gene encodes an engineered luciferase whose activity is modulated by cyclic AMP (Promega, Madison, WI, USA)[56].

Plasmids were assembled by one-pot restriction and ligation using an optimized Golden Gate protocol. DNA fragments were obtained as double-stranded synthetic DNA gBlocks (IDT), annealed oligonucleotides (IDT), or PCR products and either used directly or subcloned into a backbone with different antibiotic resistance. For each fragment, 40 fmol were added to a reaction containing 0.7 µl highly concentrated (HC) T4 ligase (Promega), 1 × ligase buffer, 0.3 µl BsaI and 1 × bovine serum albumin (New England Biolabs, Ipswich, MA, USA) in a final volume of 10 µl. The reaction was performed in a thermocycler using the following program: 1 × (37C, 5 min); 25 × (37C, 2 min; 16C; 5 min); 1 × (50C, 5 min; 80C, 5 min); hold at 4C. The reaction product (2 ul) was transformed into chemically competent E. coli (E. cloni 10G, Lucigen Corp.). DNA preparations from kanamycin-resistant clones were screened by restriction digest and verified by sequencing (Genewiz). Lentiviral plasmids C504, C505 and C512 were cloned as above, verified and then re-transformed into Stbl3 cells (New England Biolabs) for plasmid production. Liquid cultures were grown in 50 ml LB and DNA was extracted by midi-prep (Qiagen, Valencia, VA, USA).

**Mammalian cell culture.** HEK293FT cells were purchased from Life Technologies (Grand Island, NY) and cultured in 90% DMEM medium, supplemented with 2 mM glutamine, 10% fetal bovine serum (FBS), 100 units per ml penicillin, and 100 µg ml$^{-1}$ streptomycin. Cells were frozen in freezing medium composed of 50% unsupplemented DMEM, 40% FBS and 10% dimethylsulfoxide. The cells tested negative for mycoplasma contamination in the MycoAlert assay (Lonza, Walkersville, MD, USA). This cell line was chosen because of its extensive prior use and validation for lentivirus production[51], for CGRP receptor activity assays[55], and for brain implantation[57]. We performed no further authentication of the identity of the HEK293FT cell line because we obtained it from a trusted source and because functional validation of lentiviral gene transfer, of CGRP reporter performance, and of CGRP secretion as described in the main text was satisfactory.

**Lentivirus production.** HEK293FT cells were seeded into 6-well plates at 1 million cells per well and transfected using Lipofectamine 2000 (Life Technologies, Grand Island, NY) according to instructions at sub-confluence. Co-transfection of 0.5 µg pMD2.G, 1 µg psPAX2 and 1 µg of the lentiviral plasmid of interest was performed with 6.25 µl Lipofectamine 2000 reagent. Virus-containing supernatant was collected after 48 and 72 h, filtered through 0.45 µm filters, and used for infection without further concentration. Supernatants were stored at 4 °C for up to a week.

**Lentiviral infection and cell line generation.** HEK293FT cells were seeded into 24-well plates at 40,000 cells per well (final) in the presence of 4 µg ml$^{-1}$ Polybrene in 50% fresh medium and 50% viral supernatants containing between one and three different viruses. The medium was replaced with fresh viral supernatants daily for 2 days (infection with single virus) or 4 days (triple infection). Selection was performed using both antibiotic resistance and fluorescent markers for each lentivirus. Beginning on day 3 after initial infection, appropriate antibiotics (blasticidin at 10 µg ml$^{-1}$, puromycin at 1 µg ml$^{-1}$, hygromycin at 250 µg ml$^{-1}$, or combinations; all from Life Technologies) were added to the medium for selection and selection was continued until all cells expressed the appropriate fluorescent markers. Selection was then discontinued, cells were expanded and aliquots were frozen down.

**Luminescent cAMP assay for CGRP receptor activation.** The GloSensor cAMP assay[56] (Promega) was used to measure cAMP generation upon CGRP receptor activation in real time. HEK293FT reporter cells were virally transduced with an engineered luciferase whose activity is subject to fast allosteric modulation by cytosolic cAMP. We generated one HEK293FT cell line carrying only the lentiviral construct encoding the engineered luciferase (negative control) and another cell line carrying three lentiviral expression constructs, encoding the engineered luciferase and the two components of the heterodimeric CGRP receptor (CGRP reporter cell line); see Supplementary Fig. 2.

On day 0, 10,000 cells per well were seeded in 100 µl DMEM + 10% FBS in white opaque clear-bottom 96-well plates (Costar #3610, Coppell, TX). On day 1 or 2, the medium was removed from the wells and replaced with 90 µl per well of Gibco $CO_2$-independent medium (Life Technologies) + 10% FBS containing 1% v/v of cAMP GloSensor substrate stock solution (Promega). The cells were incubated in substrate-containing medium at 37 °C in 5% $CO_2$ for at least 2 h (maximally 8 h). Before the luminescence bioassay, cells were removed from the cell culture incubator and equilibrated to room temperature and atmospheric $CO_2$ for 30 min. Then, a pre-addition read was performed for 10 min to establish a baseline for luminescence. Compounds and reaction products to be tested were

quickly added in 10 μl volume per well at 10 × of the desired final concentration using a multichannel pipette. A 30 min time resolved readout of luminescence was then performed post-addition with time points every 60 or 90 s, and the time course was examined to confirm a plateau in the signal after 10–15 min, persisting at least through the 25 min time point. All further data analysis was performed on the basis of the luminescence intensity at the 15 min post-addition time point.

**Bioassay data analysis.** Data analysis was performed with a custom Python program. Unless stated otherwise, luminescence intensities at minute 15 were normalized within each microplate to 0.0 = blank signal from reporter cells with buffer added and 1.0 = signal from reporter cells with 100 nM of wild-type synthetic human alpha CGRP (Sigma) added. Triplicate wells were grouped and the mean and standard deviation for each compound concentration were plotted and used for weighted least squares regression. Dose–response curves were fit to a four-parameter Hill equation of the following form:

$$y = a + \frac{(b-a)x^n}{x^n + (10^k)^n} \tag{1}$$

$x$ is the concentration of CGRP or equivalent, $y$ is the luminescence reading, $a$ and $b$ are baseline constants, $k$ is the logarithm of the $EC_{50}$ value, and $n$ is the effective Hill coefficient. Fitting was performed by weighted nonlinear least-squares regression with the Levenberg-Marquardt algorithm as implemented in the leastsq module from the Scientific Python library. For dose–response curves that did not reach a plateau over the examined concentration range, the values of the asymptotes were constrained to $a$ = mean value of blank and $b$ = mean value of 100 nM wtCGRP and only $n$ and $EC_{50}$ were allowed to vary. For curves that never left the baseline (maximum activity < 10% of activity of 100 nM wtCGRP), no attempt was made to determine an exact value of $EC_{50}$. Standard errors for $\log(EC_{50})$ values were computed using the delta method and converted to asymptotic 95% CI (CI values noted in the text) for $EC_{50}$ and $\log(EC_{50})$.

**Reporter validation.** Forskolin, wtCGRP, and [8–37]CGRP were purchased from Sigma-Aldrich and used in bioassays as described above. To determine whether CGRP responses were specifically mediated by RAMP1/CLR, two HEK293FT cell lines were used, one expressing the Glo22F cAMP-responsive luciferase only and the other also expressing the RAMP1/CLR CGRP receptor heterodimer. All luminescence values were normalized from zero to one using baseline values obtained from buffer only or addition of 100 μM forskolin on the same respective cell line. In a separate experiment, CGRP dose–response curves were obtained in the presence of [8–37]CGRP on reporter cells expressing RAMP1 and CLR. [8–37]CGRP is an N-terminally truncated CGRP variant that acts as a competitive inhibitor of CGRP agonist activity and is expected to shift CGRP dose–response curves to the right. Here luminescence data were normalized between values obtained for buffer only and 100 nM CGRP only.

**Peptide synthesis.** CGRP-like peptides were made by solid-state synthesis at the MIT Koch Institute Biopolymers lab, oxidatively cyclized and purified by high-performance liquid chromatography. Identity and purity were confirmed by matrix-assisted laser desorption ionization-time of flight (MALDI-TOF) mass spectrometry and analytical high-performance liquid chromatography. After lyophilization, the peptides were weighed, dissolved in water or 50% dimethylsulfoxide, and quantified rigorously using a fluorescent microplate assay (FluoroProfile, Sigma-Aldrich) with CGRP from Sigma-Aldrich as a concentration standard. The peptide solutions were then adjusted to a stock concentration of 100 μM and stored at − 20 °C.

**Production and purification of GFP-CGRP fusions.** GFP-CGRP fusions were expressed from a T7 promoter upon induction with 0.4 mM isopropyl β-D-1-thiogalactopyranoside in *E. coli* BL21 cells (New England Biolabs). Cells were grown in 50 ml LB medium at 37 °C to an optical density at 600 nm of 0.4–0.6, induced, and expression was allowed to proceed for 3–5 h at 37 °C. Cell pellets were collected, lysed with BugBuster (EMD Millipore, Billerica, MA), and processed for inclusion body preparation according to the manufacturer's instructions. Inclusion bodies were dissolved in 5 ml denaturation buffer (6 M guanidinium HCl, 20 mM dithiothreitol, 50 mM Tris-HCl, pH 7.5) overnight and the absorbance at 280 nm ($A_{280}$) was measured. Extinction coefficients at 280 nm ($\varepsilon_{280}$) and molar masses (MWs) were computationally predicted from the amino-acid sequence [sensor (2): $\varepsilon_{280} = 26,980$ cm$^{-1}$ M$^{-1}$, MW = 33,174 Da; sensor (3): $\varepsilon_{280} = 25,700$ cm$^{-1}$ M$^{-1}$, MW = 32,911 Da; sensor (4): $\varepsilon_{280} = 25,700$ cm$^{-1}$ M$^{-1}$, MW = 32,780 Da]. From these values and the measured 280 nm absorbances, concentrations were then calculated and adjusted to 0.5 mg ml$^{-1}$ for refolding. Refolding was performed by dialysis of 40 ml denatured protein solution against refolding buffer (100 mM NaCl, 20 mM Tris-HCl, pH 7.5). Insoluble material was removed from the refolded solution by centrifugation (16,000 g for 20 min) and the supernatant containing soluble protein was concentrated to a volume of 3–4 ml. The fusion protein was isolated by StrepII affinity purification using 1 ml StrepTactin agarose resin (Qiagen) with refolding buffer as wash buffer. Elution buffer additionally contained 2.5 mM desthiobiotin. Fractions containing the bulk of eluted protein (as judged by green colour) were pooled, buffer exchanged

into storage buffer (identical to refolding buffer) using NAP-10 size exclusion columns (GE Healthcare), and concentrated with spin columns. $A_{280}$ values were determined and concentrations of the stock solutions were adjusted to 100 μM using predicted extinction coefficients and MWs. Purity and identity of the proteins were verified by sodium dodecylsulfate–polyacrylamide gel electrophoresis and MALDI-TOF mass spectrometry. Stock solutions were stored for up to 2 weeks at 4 °C.

**Proteolytic cleavage reactions.** Recombinant human FAP alpha expressed in Sf21 cells (400–600 ng μl$^{-1}$; Sigma-Aldrich product F7182), recombinant human CASP3 expressed in *E. coli* (230 ng μl$^{-1}$; Sigma-Aldrich product C1224), ProTEV Plus protease (5 U μl$^{-1}$; Promega product V6101) and enterokinase light chain (2 ng μl$^{-1}$; New England Biolabs product P8070) were purchased from commercial sources and used at the final concentrations indicated for each experiment. Reactions were performed by combining varying concentrations of the relevant protease and the desired concentration of the sensor peptide or fusion protein (10 × the highest desired final concentration in the bioassay) in reaction buffer (100 mM NaCl, 20 mM Tris-HCl, pH 7.5), and incubating the mixture at 37 °C for 2 h. These conditions permitted all investigated proteases to achieve near-complete cleavage, as indicated by our assays. For cleavage reactions followed by serial dilution and CGRP bioassays, 1% 3-[(3-cholamidopropyl)dimethylammonio]-1-propanesulfonate (CHAPS) was included in the reaction buffer to reduce adsorptive loss of free peptides. For cleavage mixtures used for *in vivo* injection, CHAPS was omitted. The extent of cleavage was assessed in each case by SDS–polyacrylamide gel electrophoresis for fusion proteins and by MALDI-TOF mass spectrometry for small peptide substrates.

***In vitro* bioassay for CGRP secretion.** Gelatin-coated 6-well tissue culture plates were seeded with $10^6$ HEK293FT cells per well carrying a lentiviral transgene (either C503 prepro-CGRP or C501 control) on day 1. On day 2, each well was covered by ∼ $2 × 10^6$ confluent, contact-inhibited cells. The medium was aspirated, the cell layers were washed twice with fresh medium, and 2 ml fresh medium were added. Aliquots of the culture supernatant (100 μl) were taken immediately and again after 24 h (on day 3). To quantify the CGRP-like bioactivity in the culture supernatants, 10 μl of each sample was added to bioassay wells containing 90 μl CO$_2$-independent medium, in triplicate. As a positive control and reference standard, known dilutions of synthetic wtCGRP (Sigma) were prepared in culture medium + 0.1 % CHAPS and added at the same 1:10 dilution. Culture medium alone served as the buffer-only control.

CGRP secretion (in mol per cell per day) was calculated from the CGRP bioactivity in the culture supernatant (in nM) by assuming $2 × 10^6$ in 2 ml medium, or $10^9$ cells l$^{-1}$ and, therefore, multiplying the measured concentration by $10^9$ (nM/M) × $10^9$ (cells l$^{-1}$). The required concentration of cells (in cells μl$^{-1}$) for *in vivo* detection was calculated by assuming a need for 10 nM CGRP ($10^{-8}$ mol l$^{-1}$ = $10^{-14}$ mol μl$^{-1}$) per day by the quantity of secreted CGRP (in mol per day) per cell.

**Preparation of cells for *in vivo* implantation.** HEK293FT cells carrying a C503 prepro-CGRP or a C501 control lentiviral expression cassette were seeded at 50% confluence ($7.5 × 10^6$ cells per plate) in 10 cm cell culture plates and grown to confluence overnight. The next day, the cell layer was washed twice with fresh medium, aspirated and scraped. The total volume was adjusted to 150 μl with fresh medium for a density of $10^5$ cells μl$^{-1}$ and pipetted up and down to break up clumps until a homogenous cell suspension was obtained. Dispersion and integrity of cells was confirmed by brightfield microscopy.

**Software availability.** Scientific Python is freely available at https://www.scipy.org. AFNI is a freely available at https://afni.nimh.nih.gov/afni. The customized scripts used for processing the data in this paper are available from the corresponding author upon reasonable request.

**Data availability.** Full sequences for all plasmids are provided in Supplementary Tables 1–8. The data that support the findings of this study are available from the corresponding author upon reasonable request.

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

## Acknowledgements

Project funding was provided by NIH grants R01-MH103160, R01-NS076462, BRAIN Initiative award R24-MH109081 and an MIT Simons Center for the Social Brain (SCSB) Seed Grant to A.J. A.L.S. was supported by predoctoral fellowships from the Boehringer-Ingelheim Fonds and the Friends of the McGovern Institute. A.C. was supported by the SCSB undergraduate research opportunities program. We thank Didier Trono and Eric Campeau for genetic constructs used in the research.

## Author contributions

A.L.S., A.C., and M.B. performed *in vitro* experiments. M.D. performed *in vivo* experiments. M.D., A.L.S., and A.J. designed the research and wrote the paper.

## Additional information

**Competing financial interests:** The authors declare no competing financial interests.

