## [Peer Review File · Nature Communications]

Reviewer #1

The paper covers a topic of importance and is of overall interest.

The authors appear to have done pioneering work by conceptualizing and demonstrating the possibility to variably induce vasodilation by molecular based mechanisms to enable molecular imaging via optical or MRI.

The paper unfortunately is not well structured and leaves the reader lost in the wealth of detail. Less is more, is a very appropriate characterization for the paper and how it should be improved.

Substantial credibility is compromised with naïve statements about current and alternate imaging approaches. This has to be highlighted, as the author imply the translational potential of their approach and related discoveries. While the reviewer does not disagree with that potential, the paper require a more realistic perspective. There are considerable challenges to the translation into larger animals and eventually into humans. It might be a preferable paper structure if the authors focus on either the mechanistic concept discovered, proposed and demonstrated and leave out all "comparisons" to current imaging approaches, or focus and discuss on the translational aspects, challenges and opportunities.

The discussion is overall brief and shallow, it would benefit by more rigorous discussion. The author reference possible limitations of CGRP due to complications in migraine, but should consider the potential pathophysiological aspects and implications.

While the online and methodology section is adequate, the authors should consider moving a lot more details from the main text into this section and replacing with substantial more structure and perspectives.

The authors work and concepts are exciting and of major relevance with plenty of scientific opportunities, however the paper structure and organization got lost, which is understandable due to the authors enthusiasm and appears readily fixable, but requires appropriate effort.

Reviewer #2

A. Summary of the key results

This is a vey novel study that uses the sensitivity of smooth muscle in normal vasculature to respond to CGRP (a potent vasodilator) to act as an in vivo probe to detect cellular events by blood oxygenation sensitive MRI. A variety of constructs are used to demonstrate the applications: the of effect direct injection of CGRP is studied in rat brain; the ability to detect CGRP expressing cells that had been injected is demonstrated; the ability to detect proteases that cleave an inactive CGRP complex to its active form is demonstrated.

B. Originality and interest: if not novel, please give references

This is an original study that could have wide-spread applications for studying molecular processes in vivo by MRI with unique sensitivity and specificity.

C. Data & methodology: validity of approach, quality of data, quality of presentation

The authors have done excellent work in validating the sensitivity and methodology with in vitro assays prior to in vivo studies with MRI and optical microscopy for validation.

D. Appropriate use of statistics and treatment of uncertainties

Appropriate numbers of replicates for the cell studies appear to have been performed and the n=1 and average n=5 data for MRI shows very convincing results.

E. Conclusions: robustness, validity, reliability

These initial studies with direct injections look very promising and it will be interesting to see how the method develops for pathological studies. Maybe some comment on the limitations could be made, since the response to CGRP requires the presence of functional smooth muscle, hence with disease states such as brain tumours and stroke there may be limited applications or ambiguous (the steal effect) results

F. Suggested improvements: experiments, data for possible revision

None.

G. References: appropriate credit to previous work?

Good referencing

H. Clarity and context: lucidity of abstract/summary, appropriateness of abstract, introduction and conclusions

Good methodological detail is presented within the on-line sections.

Reviewer #3

This is an interesting molecular study of a novel method to study CGRP in the brain. The MRI and cellular methodology are fine but the role of the blood-brain barrier is not considered enough. The results show elegant responses of CGRP vasodilatation when applied onto or within the parenchyma.

Page 4 line 5 in Results: There exist few studies on intracerebral arterioles and those cited are not correct.

Page 5 line 6: Was topical administration of CGRP (50 nM) given in all experiments in the same way?

Figure 2. Was CGRP applied in the same way in all experiments - topical on the cortex? The results in the illustration part c are sound regarding artery diameter and in concert with previous work (see McCulloch, PNAS 1986). At the same time the optical signal points downward to the same application. In d) the term intracranial injection of CGRP is used - is it the same topical way and dose? Or if I look on the picture there are needle holes on both sides indicating an intraparenchymal injection on one side of CGRP and of aCSF on the other side. Needs clarification! On the other side in e) the authors use L-NAME to test role of NO again with as I presume injection into the parenchyma.

As stated on page 5 line 12; CGRP responses can be studied. However CGRP does not pass the BBB hence only if the barrier is broken with the large needle into the cortex (Edvinsson, Br J Pharmacol 2007). The BBB should be evaluated with some tracer or simply checked by Evans Blue Albumin study. While the recording method is fine the way the drug is given is to my mind a flaw in the work.

Page 6 lines 7-8. L-NAME does not differentiate between types of NOS (e, n or i). It is true that CGRP dilates brain vessels via a mechanism that is unrelated to the endothelium and NO production (Edvinsson, Neurosci Lett 1985), however, did you not see an effect per se of L-NAME?

Page 12 line 16: Ho, Nature Rev Neurol 2010 might be useful for CNS actions and in particular primary headaches like migraine.

Reviewer #4

A. Summary of the key results

The paper describes a novel concept for molecular imaging which is applicable - at least in principle - to different imaging modalities. The experimental work is focused on its application in MRI. The basic idea is to use an agent, which leads to MR-visible signal changes at very low tracer concentrations. This way physiology is used to amplify signal detection. Experimental data demonstrate that changes induced by tracer concentrations much below the MR detection limit can be observed by the resulting physiological changes.

B. Originality and interest: if not novel, please give references

The idea to use some suitable physiological response to amplify the signal is innovative and ingenious and promises to open up new applications.

C. Data & methodology: validity of approach, quality of data, quality of presentation

The molecule used as tracer is calcitonin gene-related peptide (CGRP) which is a very powerful vasodilatory peptide. Three possible modes of application are described: direct observation of vasodilation mediated by CGRP, fusion of CGRP to a semi-stable blocking domain, which is then locally released by the target, and application of CGRP as a genetically encoded reporter. For all three modes of application experimental proof-of-principle studies have been performed.

It does have, however, some inherent challenges:

1.) the issue of delivery has been mentioned. For a biomedical application the agent should be applied intravenously, the challenge will then be to bring it across the vessel wall to reach its target. For a small sized peptide this is not trivial, but also not unsurmountable.

2.) a more serious inherent limitation is the lack of quantitation. The measured signal will depend on a rather complex chain of factors - blood flow to the target area, permeability of the vessels, receptor density (which is the parameter of interest), physiological response and finally response of the measurement system (MRI) to the physiological response. It will be a severe challenge to measure all relevant parameters involved in order to attach some meaningful quantitative numbers to the measurements, so it is to be expected that the range of applications will be limited to pure mapping experiments. There are a few instances where pure mapping is sufficient (e.g. fMRI), but in many other potential areas of application this will be a severe limitation.

E. Conclusions: robustness, validity, reliability

Overall the paper is very well written, experiments have been performed with serious consideration to all pertinent factors, the authors have done a very good job to provide all relevant experimental data.

F. Suggested improvements: experiments, data for possible revision

As a minor issue the evaluation used for supplementary figure one looks somewhat weird. Placing rectangular U-shaped boxes around the area of injection looks strange. The message is clear that the response decreases with distance to the injection, but it would be good to find some more physiologically meaningful way to demonstrate this.

G. References: appropriate credit to previous work?

yes

Clarity and context: lucidity of abstract/summary, appropriateness of abstract, introduction and conclusions

all very well

We thank the reviewers for their constructive comments. We believe we have been able to address the specific concerns of the reviewers in full. Our responses below are cross-referenced to changes in the manuscript, and we supply an annotated version of the main text with major changes in highlighted in red.

Reviewer 1

1. The paper covers a topic of importance and is of overall interest.

The authors appear to have done pioneering work by conceptualizing and demonstrating the possibility to variably induce vasodilation by molecular based mechanisms to enable molecular imaging via optical or MRI.

We thank the Reviewer for these kind remarks.

2. The paper unfortunately is not well structured and leaves the reader lost in the wealth of detail. Less is more, is a very appropriate characterization for the paper and how it should be improved.

We appreciate this critical feedback, and we have added more subheadings, paragraph breaks, and textual landmarks in the Results section (pp. 4-12) to better guide the reader through the experiments. We hope that these changes will enable readers to better focus on key messages and skip past details they find extraneous. We confess that we have retained the original level of experimental detail at the same time, however. We acknowledge that there is room for differences of taste regarding the level of detail included in a manuscript, but we note that none of the other three Reviewers asked for our text to be pared down, and we have attempted as well as possible to satisfy all critics.

3. Substantial credibility is compromised with naïve statements about current and alternate imaging approaches. This has to be highlighted, as the author imply the translational potential of their approach and related discoveries.

In response to this comment, we have expanded our survey of current and alternate imaging approaches in the Introduction (pp. 3-4) and tried to be more even-handed in our brief review of strengths and limitations of existing techniques. In the Discussion section, we have also substantially extended our considerations of limitations of our new technique and the future challenges of translating it (pp. 12-14).

4. While the reviewer does not disagree with that potential, the paper require a more realistic perspective. There are considerable challenges to the translation into larger animals and eventually into humans. It might be a preferable paper structure if the authors focus on either the mechanistic concept discovered, proposed and demonstrated and leave out all "comparisons" to current imaging approaches, or focus and discuss on the translational aspects, challenges and opportunities.

We address this criticism through major changes to the Discussion section. In particular, we have considerably expanded our consideration of the limitations of our imaging approach, including the need for probe delivery, the difficulty of quantification, the uncertain consequences of vascular pathology on the readouts, and the pathophysiology of CGRP itself (pp. 12-14). As part of this added discussion, we touch on several challenges likely to be encountered as part of translation to humans.

5. The discussion is overall brief and shallow, it would benefit by more rigorous discussion. The author reference possible limitations of CGRP due to complications in migraine, but should consider the potential pathophysiological aspects and implications.

We hope that the changes noted with regard to the previous point appropriately address this criticism.

6. While the online and methodology section is adequate, the authors should consider moving a lot more details from the main text into this section and replacing with substantial more structure and perspectives.

The authors work and concepts are exciting and of major relevance with plenty of scientific opportunities, however the paper structure and organization got lost, which is understandable due to the authors enthusiasm and appears readily fixable, but requires appropriate effort.

We have tried to address this comment by enhancing our Introduction and Discussion sections (points 3-4 above), and by introducing additional subheadings and breaks to guide the reader through our Results section (see point 2).

Reviewer 2

A. Summary of the key results

This is a very novel study that uses the sensitivity of smooth muscle in normal vasculature to respond to CGRP (a potent vasodilator) to act as an in vivo probe to detect cellular events by blood oxygenation sensitive MRI. A variety of constructs are used to demonstrate the applications: the effect of direct injection of CGRP is studied in rat brain; the ability to detect CGRP expressing cells that had been injected is demonstrated; the ability to detect proteases that cleave an inactive CGRP complex to its active form is demonstrated.

B. Originality and interest: if not novel, please give references

This is an original study that could have wide-spread applications for studying molecular processes in vivo by MRI with unique sensitivity and specificity.

C. Data & methodology: validity of approach, quality of data, quality of presentation

The authors have done excellent work in validating the sensitivity and methodology with in vitro assays prior to in vivo studies with MRI and optical microscopy for validation.

D. Appropriate use of statistics and treatment of uncertainties

Appropriate numbers of replicates for the cell studies appear to have been performed and the n=1 and average n=5 data for MRI shows very convincing results.

We thank the Reviewer for this positive assessment.

E. Conclusions: robustness, validity, reliability

These initial studies with direct injections look very promising and it will be interesting to see how the method develops for pathological studies. Maybe some comment on the limitations could be made, since the response to CGRP requires the presence of functional smooth muscle, hence with disease states such as brain tumours and stroke there may be limited applications or ambiguous (the steal effect) results

We address this comment in an expanded discussion of the current limitations of our technology (pp. 12-14). At the top of p. 14, we specifically touch on issues that might complicate detection of vasoactive probes or interfere with vascular reactivity required in order for the probes to function.

F. Suggested improvements: experiments, data for possible revision

None.

G. References: appropriate credit to previous work?

Good referencing

H. Clarity and context: lucidity of abstract/summary, appropriateness of abstract, introduction and conclusions

Good methodological detail is presented within the on-line sections.

We appreciate the Reviewer's input on these points.

Reviewer 3

1. This is an interesting molecular study of a novel method to study CGRP in the brain. The MRI and cellular methodology are fine but the role of the blood-brain barrier is not considered enough. The results show elegant responses of CGRP vasodilatation when applied onto or within the parenchyma.

We thank the Reviewer for a positive response. The issue of how CGRP-based probes could be delivered past the blood-brain barrier is an important one which we address with revised text on pp. 12-13.

2. Page 4 line 5 in Results: There exist few studies on intracerebral arterioles and those cited are not correct.

We thank the Reviewer for pointing out this error in our referencing. The correct reference is now cited on p. 5 of the revised manuscript [Edwards *et al.*, Calcitonin gene-related peptide

stimulates adenylate cyclase and relaxes intracerebral arterioles. *J. Pharmacol. Exp. Ther.* **257**, 1020-1024 (1991)].

3. Page 5 line 6: Was topical administration of CGRP (50 nM) give in all experiments in the same way?

Topical administration of 50 nM wtCGRP was applied in all optical imaging of the exposed cortex in the same way. Deep injections of 100 nM wtCGRP were performed for all MRI experiments. We attempt to clarify with wording changes on pp. 5-6 and in the caption to Fig. 2. We also provide a full description of the relevant methods on pp. 15-20.

4. Figure 2. Was CGRP applied in the same way in all experiments - topical on the cortex? The results in the illustration part c are sound regarding artery diameter and in concert with previous work (see McCulloch, PNAS 1986). At the same time the optical signal points downward to the same application.

CGRP was applied topically to exposed cortex in the same way in all optical imaging experiments ($n = 5$). The black trace in Fig. 2c represents the percent change in optical reflectance from a parenchymal region of interest (black rectangle in Fig. 2b). This trace veers downward during CGRP infusion because of the expected increase in microcapillary blood volume, which darkens the image in response to the vasodilator. In contrast, the red trace in Fig. 2c is a direct measure of percent dilation for an identified vessel shown in panel 2b, and this measure veers upward as expected during CGRP administration. We have rephrased caption text in the caption to Fig. 2 and in the body of the manuscript to clarify these points.

5. In d) the term intracranial injection of CGRP is used - is it the same topical way and dose? Or if I look on the picture there are needle holes on both sides indicating an intraparenchymal injection on one side of CGRP and of aCSF on the other side. Needs clarification! On the other side in e) the authors use L-NAME to test role of NO again with as I presume injection into the parenchyma.

All of the MRI experiments were performed using infusions into deep parenchymal tissue, using implanted cannulae as illustrated in Fig. 2d. This procedure differs from the topical application used in the optical experiments of Fig. 2a-c. The same intracranial parenchymal injection procedure was used to deliver CGRP in the L-NAME experiments of Fig. 2e, but the L-NAME itself was delivered by systemic, intravenous injection. We attempt to clarify these details better with edits to Fig. 2 and its caption.

6. As stated on page 5 line 12; CGRP responses can be studied. However CGRP does not pass the BBB hence only if the barrier is broken with the large needel into the cortex (Edvinsson, Br J Pharmacol 2007). The BBB should be evaluated with some tracer or simply checked by Evans Blue Albumin study. While the recording method is fine the way the drug is given is to my mind a flaw in the work.

Both the topical and deep parenchymal injections of Fig. 2 were performed by using invasive measures to circumvent the BBB, in the first case by breaching the meninges, and in the second

by inserting injection cannulae. There can be no doubt that the BBB was compromised in both conditions. To extend use of the vasoactive molecular imaging technique to truly noninvasive applications, it will indeed be a challenge to establish an effective means to deliver CGRP-based probes more passively past the BBB. We discuss this issue on pp. 12-13 of the revised Discussion.

7. Page 6 lines 7-8. L-NAME does not differentiate between types of NOS (e, n or i). It is true that CGRP dilates brain vessels via a mechanism that is unrelated to the endothelium and NO production (Edvinsson, Neurosci Lett 1985), however, did you not see an effect per se of L-NAME?

We reexamined the data in response to the Reviewer's question and do in fact observe a small increase in the baseline MRI signal (< 5%) during the first five minutes after L-NAME injection. By altering the dynamic range for CGRP-mediated effects, this change could in principle have contributed to the slight reduction in CGRP evoked signal after L-NAME injection. The difference was not found to be significant for our sample, however ($n = 4$). We note this point on p. 7 of the revised text.

8. Page 12 line 16: Ho, Nature Rev Neurol 2010 might be useful for CNS actions and in particular primary headaches like migraine.

We thank the Reviewer for pointing us to this reference, which we have now cited on p. 14 of the revised manuscript.

Reviewer 4

A. Summary of the key results

The paper describes a novel concept for molecular imaging which is applicable - at least in principle - to different imaging modalities. The experimental work is focused on its application in MRI. The basic idea is to use an agent, which leads to MR-visible signal changes at very low tracer concentrations. This way physiology is used to amplify signal detection. Experimental data demonstrate that changes induced by tracer concentrations much below the MR detection limit can be observed by the resulting physiological changes.

B. Originality and interest: if not novel, please give references

The idea to use some suitable physiological response to amplify the signal is innovative and ingenious and promises to open up new applications.

We thank the Reviewer for this enthusiastic reception.

C. Data & methodology: validity of approach, quality of data, quality of presentation

The molecule used as tracer is calcitonin gene-related peptide (CGRP) which is a very powerful vasodilatory peptide. Three possible modes of application are described: direct observation of vasodilation mediated by CGRP, fusion of CGRP to a semi-stable blocking domain, which

is then locally released by the target, and application of GGRP as a genetically encoded reporter. For all three modes of application experimental proof-of-principle studies have been performed.

It does have, however, some inherent challenges:

1.) the issue of delivery has been mentioned. For a biomedical application the agent should be applied intravenously, the challenge will then be to bring it across the vessel wall to reach its target. For a small sized peptide this is not trivial, but also not unsurmountable.

We completely agree with the Reviewer that probe delivery to the brain is an important challenge. In the revised manuscript, we explicitly note the importance of this issue to translational applications (p. 13). We also discuss the possibility of using receptor-mediated transcytosis as a means for delivering the probes noninvasively in the future.

2.) a more serious inherent limitation is the lack of quantitation. The measured signal will depend on a rather complex chain of factors - blood flow to the target area, permeability of the vessels, receptor density (which is the parameter of interest), physiological response and finally response of the measurement system (MRI) to the physiological response. It will be a severe challenge to measure all relevant parameters involved in order to attach some meaningful quantitative numbers to the measurements, so it is to be expected that the range of applications will be limited to pure mapping experiments. There are a few instances where pure mapping is sufficient (e.g. fMRI), but in many other potential areas of application this will be a severe limitation.

Yes, the semi-quantitative nature of the molecular imaging readout we introduce here does represent a limitation of the technique. In the revised manuscript, we include expanded text about this and other current limitations (pp. 12-14). The problem of quantification can be avoided in some cases. For instance, a number of mapping-style experiments may not involve quantification at all; these include cell tracking (*cf.* Fig. 4), as well as potential applications such as monitoring gene therapy, infection, and metastasis. In applications where some level of quantification is desirable, techniques related to the “calibrated BOLD” procedures used in some functional MRI brain imaging experiments should be applicable. These topics are discussed in the revised text.

E. Conclusions: robustness, validity, reliability

Overall the paper is very well written, experiments have been performed with serious consideration to all pertinent factors, the authors have done a very good job to provide all relevant experimental data.

We thank the Reviewer for this appreciation.

F. Suggested improvements: experiments, data for possible revision

As a minor issue the evaluation used for supplementary figure one looks somewhat weird. Placing rectangular U-shaped boxes around the area of injection looks strange. The message is clear that the response decreases with distance to the injection, but it would be good to find some more physiologically meaningful way to demonstrate this.

We have revised Supplementary Fig. 1 by using semicircular ROIs (within the limits of voxel resolution) that correspond to discrete distances from the cannula tip more directly.

G. References: appropriate credit to previous work?

yes

Clarity and context: lucidity of abstract/summary, appropriateness of abstract, introduction and conclusions

all very well

We thank the Reviewer once again.

Reviewer #1:

The authors have taken the combined feedback of the reviewers and appropriately addressed the concerns. The paper has been substantially improved and the messages clarified. The readability could still be improved at the time of editorial type setting, again sometimes less is more.

There are no more areas of concern identifiable.

The overall tone is now appropriate for this visionary approach.

Reviewer #2:

The authors have satisfactorily addressed my comments on the first submission. This is a highly novel molecular imaging methodology for amplifying low concentration molecular interactions via hemodynamic effects. The paper presents convincing results in good detail, with considered discussion of limitations and future directions. This work promises to open up a new and exciting area of research and clinical applications.

Reviewer #3:

The changes performed by the authors have improved the manuscript well.

Reviewer #4:

The author has done an excellent job to appropriately address the issues raised in my previous report.